# FBG Interrogator Using a Dispersive Waveguide Chip and a CMOS Camera

**DOI:** 10.3390/mi15101206

**Published:** 2024-09-29

**Authors:** Zhenming Ding, Qing Chang, Zeyu Deng, Shijie Ke, Xinhong Jiang, Ziyang Zhang

**Affiliations:** 1Laboratory of Photonic Integration, School of Engineering, Westlake University, 18 Shilongshan Road, Hangzhou 310024, China; dingzhenming@westlake.edu.cn (Z.D.); dengzeyu@westlake.edu.cn (Z.D.); keshijie@westlake.edu.cn (S.K.); jiangxinhong@westlake.edu.cn (X.J.); 2Institute of Advanced Technology, Westlake Institute for Advanced Study, 18 Shilongshan Road, Hangzhou 310024, China; 3Zhejiang Engineering Research Center of Intelligent Media, Communication University of Zhejiang, Hangzhou 310018, China

**Keywords:** fiber Bragg gratings, waveguide spectral lens, FBG interrogator, integrated optics

## Abstract

Optical sensors using fiber Bragg gratings (FBGs) have become an alternative to traditional electronic sensors thanks to their immunity against electromagnetic interference, their applicability in harsh environments, and other advantages. However, the complexity and high cost of the FBG interrogation systems pose a challenge for the wide deployment of such sensors. Herein, we present a clean and cost-effective method for interrogating an FBG temperature sensor using a micro-chip called the waveguide spectral lens (WSL) and a standard CMOS camera. This interrogation system can project the FBG transmission spectrum onto the camera without any free-space optical components. Based on this system, an FBG temperature sensor is developed, and the results show good agreement with a commercial optical spectrum analyzer (OSA), with the respective wavelength-temperature sensitivity measured as 6.33 pm/°C for the WSL camera system and 6.32 pm/°C for the commercial OSA. Direct data processing on the WSL camera system translates this sensitivity to 0.44 μm/°C in relation to the absolute spatial shift of the FBG spectra on the camera. Furthermore, a deep neural network is developed to train the spectral dataset, achieving a temperature resolution of 0.1 °C from 60 °C to 120 °C, while direct processing on the valley/dark line detection yields a resolution of 7.84 °C. The proposed hardware and the data processing method may lead to the development of a compact, practical, and low-cost FBG interrogator.

## 1. Introduction

Optical sensors using fiber Bragg gratings (FBGs) have been subject to extensive research for decades, as these fiber-based sensors are flexible, lightweight, resistant to harsh environments, electrically passive, and immune to electromagnetic interferences. Therefore, such sensors are often deployed in industrial sectors under critical or extreme conditions, such as oil and gas fields, power plants, and bridges, and also as wearable devices [1,2,3,4,5]. Fundamentally, an FBG requires the periodic modulation of the refractive index along the fiber core, which results in a reflection peak at the designed wavelength λ_B_. External disturbances to the fiber, e.g., temperature change or mechanical strain, cause λ_B_ to shift. The interrogation of the sensor relies essentially on the accurate determination of this wavelength shift.

The FBG interrogators can be developed using different technologies. One type re-lies on an optical spectrum analyzer (OSA) to pinpoint the central wavelength positions. Some require a dispersive optical element and a charge-coupled device (CCD) camera so that different spectral lines are mapped to different locations on the detector plane [6]. The others adopt a wavelength sweeping laser and a tunable filter to trace the wavelength actively [7]. The method with an OSA is typically associated with high costs and limited portability, making it primarily suitable for precise measurement in labs or stable environments [8]. On the other hand, the spectrometer and CCD-integrated system demonstrates advantages in terms of their compact size, robustness, and fast acquisition rate. Nevertheless, the integrated system remains complex, requiring multiple free-space optical components such as slits, dispersive elements, and lenses. These components also need to be aligned precisely and secured in housing [7]. The sweeping laser and filter system can meet the demand for high resolution and a fast scan rate, but the cost of the tunable laser component remains high [9].

Recently, the fast advancement of photonic integration technology has led to the development of low-cost miniaturized FBG interrogators, including those based on unbalanced Mach–Zehnder interferometers [10], micro-ring resonators [11], and arrayed waveguide gratings (AWGs) [12]. Among these technologies, AWGs have gained popularity because of the high spectral resolution they can provide and their capability to interrogate multiple FBGs with a single chip [13]. The dispersed light from an AWG can be detected using photodiodes (PDs) through hybrid [14,15] or monolithic integration [16,17]. In hybrid integration, an alignment accuracy on the micrometer level is often required to place the PD array on the AWG chip facet. In monolithic integration, Ge or InGaAs PDs can be integrated on a single chip following the AWG structure without a separate off-chip attachment step. However, both methods require customized electronic circuits for relaying and processing the photocurrent. The nonstandard, off-the-shelf electronic components can increase the system cost, but also pose challenges in the reliability of the entire optoelectronic system. Therefore, it is of great interest to develop lower-cost, practical, and robust technology for FBG interrogator systems with less or better no free-space optical components and with only standard electronics, such as a commercial camera.

Previously, we proposed a waveguide-based planar spectrometer called waveguide spectral lens (WSL) [18]. The major advantage in comparison to an AWG is that a WSL can directly focus the separated spectral lines on a camera placed at the designed distance without any free-space elements. We have demonstrated the spectral analysis applications in the near infrared region (NIR, around 1550 nm) and have conducted solar spectrum measurements in the visible–near-infrared region (VNIR) below 1.1 μm [19,20].

In this work, we explore the potential to develop a practical WSL-based FBG interrogator for temperature sensing, with a minimal number of bulky free-space optical elements. Furthermore, as the wavelength is moved below 1.1 μm, only a standard CMOS camera is sufficient, eliminating the need for an expensive InGaAsP CCD camera. Last but not least, the FBG is designed to provide a peak at 830 nm, allowing a low-cost light emitting diode (LED) to be applied at this wavelength, further reducing the system cost. On the software side, to decode the temperature precisely without resorting to highly expensive temperature reference equipment, a deep neural network (DNN)-based algorithm is developed to facilitate the analysis. The results indicate that the recognition resolution for temperature reaches 0.1 °C using only a standard hotplate as the temperature controller in the range of 60 °C to 120 °C. This work may serve as a proof-of-concept for the development of cost-effective, compact, and portable FBG interrogators that enable on-site detection and analysis.

## 2. Principle and Device Design

### 2.1. Interrogation System

The proposed FBG interrogation scheme is shown in Figure 1. A superluminescent light emitting diode (SLED) with a central wavelength of 840 nm is fiber-connected to the FBG after passing through an isolator (ISO). The commercial FBG has a central wavelength of around 830 nm, a spectral width measured at the full width at a half maximum (FWHM) of 0.3 nm, and a peak reflectivity of approximately 70%. The fiber is butt-joint-coupled and secured to the WSL chip via a standard procedure as used in commercial AWGs.

The WSL chip consists of an input waveguide, a horizontal beam broadening area (BBA), and a waveguide array. The input waveguide is a single-mode waveguide that guides light to the slab region as BBA, where the beam diverges horizontally but remains confined in the vertical dimension. At the output of the BBA, a number of waveguides are placed with equal spacing along the periphery of the slab and have equal distance to the center of the front facet. Thus, the broadened wavefront of the input light is collected by the waveguides with approximately the same phase. The lengths of the waveguides are varied to control the phase for the grating effect, as well as the beam-focusing functions. The dispersion function is achieved by introducing a fixed length difference between adjacent waveguides, and additional length differences are added based on the phase modulation of a convex lens for the focusing function. The output waveguides are cut open along a line perpendicular to the equally spaced parallel waveguides. When emitting in free space, the CMOS camera positioned at the designed distance captures the dispersed light that is already focused through a modulated wavefront. Finally, the images are collected and stored in a computer for subsequent analysis.

In the interrogation system, the WSL can operate in the NIR or VNIR, covering various FBGs’ central wavelength ranges. Additionally, by flexibly designing the geometric layout of the waveguide array (e.g., using a three-arc structure [19]) and adjusting the number of waveguides in the array, it is possible to achieve a broad free spectral range (FSR), which has potential applications in multiple FBGs’ demodulation or implement high-resolution WSLs for precise FBG interrogation. Furthermore, the WSL can also be packaged in a tube and mounted to the camera with a standard C-Mount threading, leading to a handheld device for on-site measurement [19].

The WSL is fabricated on a low-cost polymer waveguide platform, and a simple fabrication process only involving standard lithography (MA6, SUSS MicroTec Co., Ltd., Garching, Germany) is sufficient. Commercial polymer materials (WIR30-RI series, ChemOptics Co., Ltd., Daejeon, Republic of Korea) with a low propagation loss at 830 nm are adopted in this work. The refractive indexes of the core and cladding materials are 1.466 and 1.45, respectively.

### 2.2. WSL Design

The critical parts in the WSL design are illustrated in Figure 2. The input waveguide is placed at the center of the BBA. The cross section of the waveguide core is 2.5 μm × 2.5 μm, as shown in the inset (I) of Figure 2. The estimated coupling loss with the single-mode fiber (S630-HP, Nufern Co., Ltd., East Granby, CT, USA) is below 1 dB and the matched effective index also results in negligible back reflection. The designed waveguide has a single-mode cutoff at 795 nm, and the shortest detectable wavelength is approximately 600 nm, where the first order mode (odd symmetrical mode) appears. However, the fiber remains single mode (symmetric, Gaussian-like) at 600 nm. For symmetric and center-alignment coupling from the fiber, the first-order mode in the input waveguide cannot be generated efficiently. Similarly, at the output facet of the BBA, the first-order modes in the waveguides have low coupling efficiency with the output light field of the BBA. Thus, the waveguide behaves in the single-mode domain for the VNIR.

As shown in Figure 2, the starting points of the waveguides in the array are placed along the perimeter of the BBA in order to tap out the light wavefront with equal phase. Each waveguide path in the array is composed of an input taper segment (S_1_), in order to vary the waveguide width smoothly to that of a single-mode waveguide, connected with a straight segment (S_2_), an arc segment (S_3_), another straight segment (S_4_), and an output taper segment (S_5_). The length difference between adjacent waveguides can be flexibly adjusted and is determined by the segments S_2_, S_3_, and S_4_, as the input and output taper lengths of all the waveguides are identical. The length differences for dispersion and focusing functions are designed based on the FSR and focal length [19]. The number of waveguides (N) in the array is determined by the required FWHM, which can be approximated by FWHM ≈ FSR/(1.13 N) [19].

Furthermore, the free-space diffraction loss caused by unwanted diffraction orders is reduced by the output waveguide design. Similar to the classic multi-slit interference, the unwanted diffraction orders can be suppressed by reducing the width of the far-field envelope and increasing the angular separation between the adjacent diffraction peaks (Δθ = 2arcsin(λ/(2d))) [21]. Here, d represents the waveguide spacing on the chip facet, as illustrated in the inset (II) of Figure 2. The width of the far-field envelope can be reduced by increasing the waveguide width on the chip facet, as the envelope is determined by the diffraction pattern of a single waveguide. To achieve this, waveguide tapers are introduced on the output side of the waveguides, as shown in the inset (II) of Figure 2.

According to the design principles above, WSL is designed for FBG interrogation. The diffraction order calculation formula is given by:m = [λ_0_/FSR × n_e_/n_g_],(1)
where λ_0_ is the central wavelength of WSL, and n_e_ and n_g_ are the effective refractive index and group refractive index of the waveguide, respectively. The square bracket indicates rounding to the nearest integer [22]. The formula for calculating the length difference of the dispersion function is ΔL = mλ_0_/n_e_ [23]. The focal length (f) of the WSL is set to 3 cm to achieve a moderate beam broadening in the vertical dimension, limited by the size/height of the detector. To minimize free-space diffraction losses and avoid high crosstalk, the width and pitch of the output waveguide at the chip facet are set to d = 10 μm. The number of waveguides (N) in the array is set to 100. The other parameters of the WSL are λ_0_ = 819.7 nm, m = 21, ΔL = 11.81 μm, FSR = 38.8 nm, and FWHM = 0.39 nm.

### 2.3. Principle of WSL-Based FBG Interrogation

The structure and principle of the FBG are illustrated in Figure 1. When light from SLED injects into FBG, the FBG reflects narrowband spectral components, while other wavelengths are transmitted. In an FBG, the central wavelength λ_B_ is given by [24]:λ_B_ = 2n_eff_Λ,(2)
where n_eff_ is the effective refractive index of the fiber and Λ is the grating period. When the FBG is affected by temperature, the shift in central wavelength (∆λ) can be expressed as [25]:∆λ = λ_B_⋅(α + ξ) ·∆T,(3)
where α and ξ are the thermal expansion coefficient and the thermo-optic coefficient of optical fiber, respectively. ∆T is the temperature variation.

As illustrated in Figure 1, the spectrum of SLED is represented as I(λ), while the filter function of the FBG is denoted as T(λ_B_). The transmitted spectrum through the FBG is I_g_(λ), where I_g_(λ) = I(λ)T(λ_B_). When the FBG experiences a temperature-induced shift in its central wavelength λ_B_, the position of the maximum dip in the I_g_(λ) also changes accordingly. The spectral response of WSL is W_ij_(λ). When I_g_(λ) enters the WSL, it is dispersed and focused onto the CMOS camera. The detected signal S_ij_ of the pixel at row i and column j can be described as the following equation [26]:(4)$$S_{\mathrm{ij}}=\int_{\lambda_{1}}^{\lambda_{2}}I\left(\lambda \right)T\left(\lambda_{B}\right)W_{\mathrm{ij}}\left(\lambda \right)\eta \left(\lambda \right)d\lambda ,i=1,2,3,\ldots \;m;j=1,2,\;3,\ldots \;n,$$
where m and n represent the number of pixels in the rows and columns of the CMOS camera, respectively. λ_1_ and λ_2_ are the start and stop wavelengths within one FSR of WSL. W_ij_(λ) means that the wavelengths dispersed by WSL are imaged on different columns of the detector along the horizontal direction of the camera and along the vertical direction of the camera, the light field follows a Gaussian distribution. η(λ) is the spectral responsivity of the commercial CMOS camera.

After the spectrum is recorded by the camera, the photon counts from each column of pixels occupied by a FSR of the WSL are summed up. The sum of pixel values in each column, V_j_, is given by
(5)$$V_{j}=\sum_{i=1}^{m}P_{\mathrm{ij}},i=1,2,3,\ldots \;m;j=1,2,3,\ldots \;n,$$
where P_ij_ denotes the pixel value at row i and column j determined by S_ij_. The position with the minimum V_j_ corresponds to the location of the maximum dip in I_g_(λ), which is the position of the FBG central wavelength λ_B_. Therefore, temperature information can be demodulated by analyzing changes in the position of this minimum V_j_.

Additionally, as described in Section 3.1, wavelength calibration employs light sources with known spectral lines to establish calibration lines/curves, mapping the pixel positions on the camera to the wavelengths. Thus, the captured image can be analyzed to extract the spectrum of the I_g_(λ) as plotted in Figure 1.

## 3. Experiment

### 3.1. WSL Fabrication and Measurement

The designed WSL is fabricated on a 4 inch silicon wafer with a standard process. The fabrication and testing processes of the polymer waveguides have been covered in our previous works [19]. The bottom cladding layer is first spin-coated onto the silicon wafer, followed by the spin-coating of a 2.5 µm-thick core layer. The device is then patterned through photolithography and inductive coupled plasma (ICP) treatment. After spin-coating and curing the top cladding layer, the wafer is sliced into bars/chips using standard sawing equipment, which are then characterized. The facets of the chips are not polished. Figure 3a shows a photo of the fabricated devices. The chip contains three WSLs with identical design parameters to efficiently use the space. The WSL located at the bottom is chosen for this study.

The waveguides are taken from the same wafer as the solar spectrum measurement in our previous work [20]. We have performed a standard cutback measurement using three sets of straight waveguides with different lengths (0.5 cm, 1 cm, and 2.5 cm) to extract the propagation loss (slope) and the coupling loss with the single mode fiber (intercept). It has been found that the fiber–chip coupling loss and the propagation loss are ~0.68 dB/facet and ~0.88 dB/cm at 824.6 nm, respectively.

Before utilizing the WSL for FBG sensing demodulation, the FWHM and FSR of the WSL are measured by wavelength calibration. In our previous work, we have elaborated on the principles and methods of wavelength calibration [20]. Similar to our previous work, the wavelength calibration of the WSL was achieved using three laser diodes (LDs). Prior to wavelength calibration, the image tilt was calibrated. The camera used in this experiment has a resolution of 2048 × 2448 pixels. During the calibration process, the tilt angle of the chip or the camera was adjusted, and the spectrum lines of an LD captured at various angles were measured. The positions of the maximum pixel values in the first and 2048th rows of the image were identified. The image was considered leveled when the maximum pixel values in row 1 and row 2048 were aligned in the same column. To ensure the accuracy of wavelength calibration, a beam splitter was used to divide the LD light into two paths: one directed to the OSA (AQ6370C, Yokogawa Electric Co., Ltd., Tokyo, Japan) for monitoring the central wavelength of the LD, and the other directed to the WSL for wavelength calibration. As shown in Figure 3b, the spectra lines of the three LDs were captured, and the central wavelengths of the three LDs, as monitored by the OSA, were 779.092 nm, 802.528 nm, and 824.616 nm, respectively. The normalized intensity distribution and the wavelength calibration results are shown in Figure 3c,d, respectively. The measured FSR and FWHM are 38.5 nm and 0.43 nm at 824.6 nm, respectively. To compare, the simulated FSR and FWHM are 39 nm and 0.39 nm at 824.6 nm, respectively. As shown in Figure 3d, there is a linear relation between the pixel position of the spectrum and the wavelength, which we have explained in detail in our previous work [20].

The differences in the FWHM and FSR between the simulation and experiment can be attributed to inaccuracies from setting the camera position/focal length, the variations in the waveguide effective index due to fabrication tolerance (random phase errors), the transmission loss imbalance of the waveguides, and the discretization of the far-field intensity distribution by the camera pixels.

### 3.2. Data Collection

In order to validate the proposed WSL-based interrogator for FBG interrogation, the FBG is put on the surface of the hotplate with no mechanical stress according to the interrogation system layout shown in Figure 1. The FBG is heated from 60 °C to 120 °C. This range is particularly important for applications, such monitoring lithium-ion batteries’ (LIBs) safety [27]. Specifically, thermal runaways in LIBs, which can lead to catastrophic failures, typically begin within this temperature range due to the exothermic decomposition of the solid–electrolyte interphase (SEI) at the anodes. A commercial OSA is used to detect the change in the central wavelength of FBG for comparison, while a WSL is used to obtain the transmission spectral lines, and the data are recorded at an interval of 10 °C. Spectral images are captured using a CMOS camera (2048 × 2448 pixels, MV-CA050-10GM, Hangzhou Hikvision Digital Technology Co., Ltd., Hangzhou, China).

## 4. Data Processing and Results

Based on the wavelength calibration results illustrated in Figure 3c,d, the spectral images acquired by the WSL-based interrogator are mapped from pixel coordinates to wavelength coordinates. This mapping enables the accurate determination of changes in the central wavelength of the FBG. Figure 4a shows a comparison between the Δ*λ_OSA_* obtained in the OSA and the Δ*λ_WSL_* obtained by the WSL-based interrogator.

Figure 4a shows that the central wavelength shift measured using the proposed interrogator exhibits a linear relation with the temperature variations, as demonstrated by a fitting equation with a linearity exceeding 0.98. The measured temperature sensitivity of the FBG is 6.33 pm/°C, corresponding to the slope of the fitting curve. For comparison, the temperature sensitivity measured using a commercial spectrometer is 6.32 pm/°C. The close agreement between the results from the demodulation system and the commercial spectrometer underscores the high accuracy of the proposed demodulation system.

To simplify the data processing, as described in Section 2.3, the temperature information can be directly demodulated by analyzing the changes in the pixel position of the minimum V_j_, which corresponds to the spectral line of the FBG center wavelength on the camera. Figure 4b shows the linear relation between the temperature and the pixel position shift of the spectral line (Δ*P*), with a sensitivity of −0.44 μm/°C, meaning that for every degree Celsius, the spectral line shifts by 0.44 μm on the camera. The negative sign indicates that as the wavelength increases, the spectral lines shift towards a direction where the number of pixel columns decreases, as illustrated in Figure 3b.

Analyzing the pixel position shift of the spectral lines provides a sensitivity of −0.44 μm/°C, indicating a clear linear relation between the temperature variations and the spectral shifts, as demonstrated in Figure 4b. In the direct analysis process, the spectral shifts are converted to the displacement change of the valley point of the dark line. However, this displacement should be larger than the width of one pixel on the CMOS camera so that the temperature variations can be distinguished. In our experiments, the CMOS camera has a pixel width of 3.45 μm and the sensitivity is measured as −0.44 μm/°C. The theoretical maximum temperature resolution can only achieve 7.84 °C. Furthermore, the wavelength resolution of the OSA is 20 pm and the measured temperature sensitivity of the FBG sensor is 6.32 pm/°C. The potential maximum temperature resolution that the OSA can achieve is calculated as 3.16 °C. To further improve the resolving ability, we employ DNN-based algorithms to process spectral image data and demodulate temperature information. Within a single pixel, as the dark line shifts from right to left, the intensity changes accordingly. This variation is interpreted as gradient information, reflecting changes in intensity values corresponding to temperature shifts. The DNN-based algorithm analyzes this gradient information by considering the valley point location in each of the 20 pixels and the relative changes between adjacent pixels across the entire 20-pixel range.

The FBG is heated from 60 °C to 120 °C with an interval of 0.1 °C, and the spectral images are captured using the CMOS camera. A total of 601 spectral images are collected for the DNN-based algorithm. The spectral image data need to be preprocessed before feeding into the DNN. As shown in Figure 5a, the Gaussian filtering is firstly applied on the image to remove high-frequency noise and enhance the contrast of the spectrum line [28]. To ensure the temperature range from 60 °C to 120 °C fully encompasses the FBG transmission spectrum shift on the camera, we hence select a rectangular region of 2048 × 20 pixels and sum each column to construct a 20 × 1 vector. The vector is determined as the input of the neural network, and the corresponding temperature is selected as the output. The structure of the neural network is also illustrated in Figure 5a; the number of the layers is set to four and the neuron number in each layer is 20, 40, 80, and 1, respectively. Architecture with more layers also works, but requires more computation, and risks running into overfitting problems, while fewer layers would lead to large errors in the training results and make it difficult for the network to converge.

A total of 601 spectral images are available for deep learning. Among these images, 60% are randomly selected for training, 20% for validation, and the remaining 20% are used for testing. The validation data were used during the training process to finetune the parameters in the network and prevent overfitting, while the testing data are only used after the model training was completed. Figure 5b displays the mean squared error (MSE) values during the training process. The loss drops below 2.4 × 10^−2^ after only 10 epochs, demonstrating that the constructed network effectively captures the relation between the input images and the corresponding temperatures. Figure 5c shows the actual temperatures alongside their corresponding predicted values generated by the trained DNN model. The blue reference line (y = x) serves as a visual guide for the ground truth. The results have proven the good performance of the DNN model and a temperature resolution of 0.1 °C is reached.

**Figure 5 micromachines-15-01206-f005:**
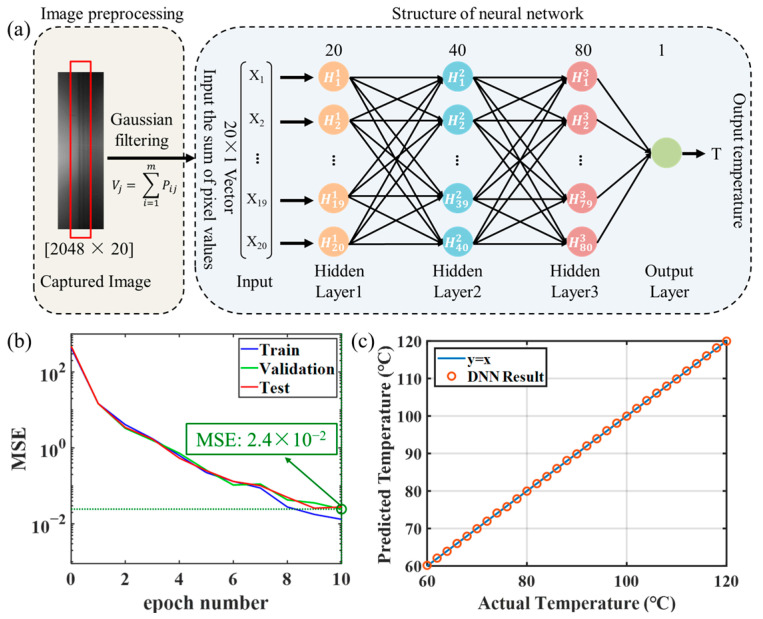
(**a**) Image preprocessing and architecture of the neural network. (**b**) The loss variation in DNN training process. MSE: mean squared error. (**c**) Scatter plot of actual temperature vs. temperature predicted by DNN.

## 5. Discussion

In our current technology, we utilize a DNN-based algorithm to enhance the resolution of the temperature on the software side. There are other ways to improve temperature sensitivity on the hardware side. Firstly, employing more sensitive FBGs can enhance the system’s temperature resolution. Currently, we use a bare FBG as the temperature sensing unit, which has limited intrinsic temperature sensitivity. Flavio Esposito et al. have developed a special FBG component, in which the FBG is encapsulated within a double-layered steel tube [29]. They reported temperature sensing ranging from 5 °C to 50 °C with a temperature sensitivity of 28.9 pm/°C, which is about three times that of a standard FBG.

Secondly, the sensitivity of the WSL-based demodulator can be enhanced by increasing the focal length of the WSL. The lateral spread-out of the far field on the camera for the wavelengths within one FSR is defined as Δw, which should be narrower than the width of the camera sensor. Δw can be approximately calculated as [21]
Δw = 2f tan(arcsin(λ/2d)),(6)
where f is the focal length of the WSL and d is the spacing between the waveguides. According to Formula (6), a larger f leads to a larger Δw, so that the wavelengths within one FSR can occupy more pixels in the lateral direction of the camera. As a result, the wavelength range corresponding to each pixel is reduced, enabling the WSL-based interrogator to detect smaller variations in wavelength. Finally, the utilization of cameras with smaller pixel dimensions can also enhance the resolution.

## 6. Conclusions

In this paper, we have proposed and demonstrated a cost-effective interrogator system for an FBG temperature sensor based on a WSL chip and a CMOS camera. The WSL can not only separate the wavelength spatially, but also focus them on a camera placed at a chosen distance without free-space optical components. Based on this technology, an FBG temperature sensor system is developed, and the results show good agreement with a commercial OSA. The wavelength temperature sensitivity for the WSL camera system and the commercial OSA are measured as 6.33 pm/°C and 6.32 pm/°C, respectively. The direct data processing on the WSL camera system translates this sensitivity to 0.44 μm/°C in relation to the image shift on the camera. However, the direct data processing using only the valley/dark lines results in a very coarse temperature resolution of 7.84 °C. To further improve the resolving ability, an DNN-based algorithm is applied to analyze the entire spectra. The temperature resolution is hence greatly improved to 0.1 °C from 60 °C to 120 °C using a standard hotplate because the DNN considers both the valley point of the dark line and the relation between the adjacent pixels in a wide range. Prospects for further improvement on the temperature sensitivity are given at the end. This study demonstrates a viable approach for developing low-cost, compact, and portable FBG interrogators capable of on-site detection and investigation.

## Figures and Tables

**Figure 1 micromachines-15-01206-f001:**
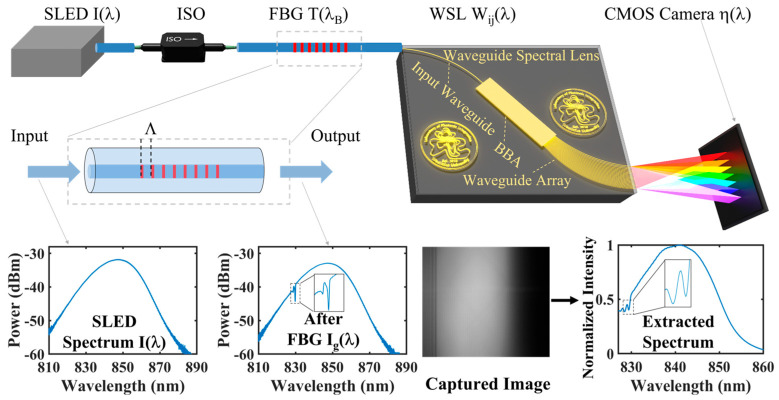
Schematic diagram of the FBG interrogation system based on a WSL. SLED: superluminescent light emitting diode; ISO: isolator; FBG: fiber Bragg grating; WSL: waveguide spectral lens; CMOS: complementary metal oxide semiconductor.

**Figure 2 micromachines-15-01206-f002:**
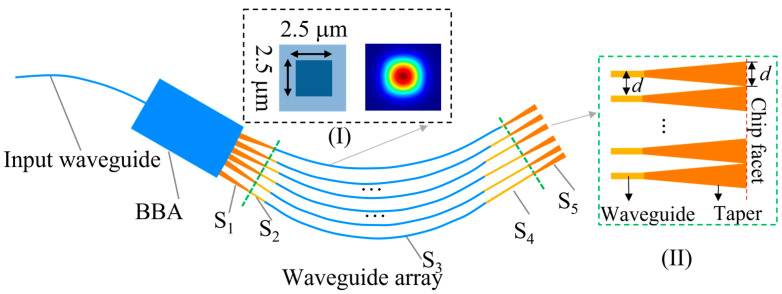
Schematic of the WSL. Inset (I) shows the cross section of the waveguide. Inset (II) shows the waveguides with output tapers designed for reducing the free-space diffraction loss. BBA: beam broadening area.

**Figure 3 micromachines-15-01206-f003:**
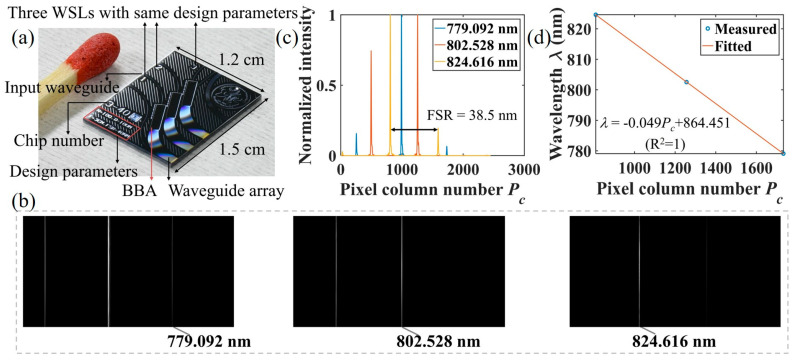
(**a**) Photo of the fabricated devices and a matchstick for size comparison. Bottom WSL is used for the FBG interrogation. Wavelength calibration of WSL: (**b**) captured spectral lines at 779.092 nm, 802.528 nm, and 824.616 nm, respectively. (**c**) Normalized intensity distributions of the spectral lines. (**d**) Wavelength calibration result.

**Figure 4 micromachines-15-01206-f004:**
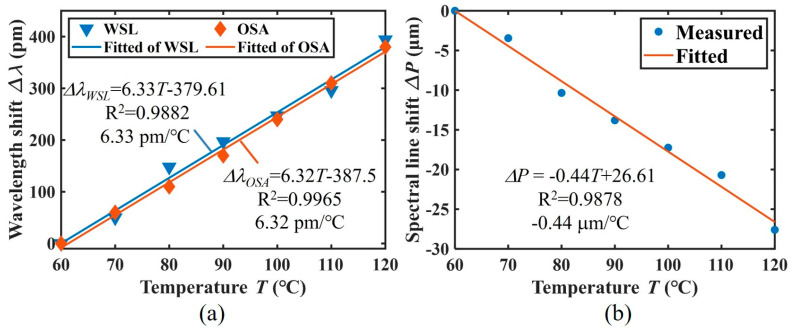
(**a**) Comparison between Δλ_B_ of the sensing FBG obtained from the OSA and from the WSL-based interrogator. (**b**) The relation between the spectral line shift and temperature.

## Data Availability

The data presented in this study are available from the corresponding author upon reasonable request.

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
