# Peer review of "FBG Interrogator Using a Dispersive Waveguide Chip and a CMOS Camera"

_micromachines, 2024, doi:10.3390/mi15101206_

Round 1

Reviewer 1 Report

Comments and Suggestions for Authors

The paper contains an application of a previous published sensing system to the sensing of temperature. Also a convolutional network (CN) is used to improve the detection resolution. The Cn seems to be the main novelty but is not treated in detail.

I have two main concerns: 1. the working of the CN is not adequately described. It generates an improvement of a factor of 70 which needs more justification. 2. The 2d nature of the (image) appears to be neglected completely despite the fact that a small tilt would allow for sub-pixel resolution. However this appears to be unused for the CN, where only a column average is inserted as input.

The CN appear to be quite large compared to the small input vector (20 degrees of freedom) and data-set (only 400 for training). It is not clear what distinguishes the validation from the testing data.

Comments on the Quality of English Language

This is adequate

Author Response

Response Letter to Reviewers

We would like to express our deep gratitude to the reviewers for reading our manuscript carefully as well as for the efforts they have spent in coming up with insightful comments and suggestions to improve the quality of the manuscript. We have tried our best to reply to the comments and made changes in the revised manuscript accordingly.

To make the text clearer and more readable, the original comments from the reviewers and editors are styled italic, our replies are styled partially in bold, and further indications of changes in the manuscript are underlined. The line, paragraph and page numbers are referred to the resubmitted manuscript.

-------------------------------------------------------------------------------------------------------

Reviewer 1

Comments and Suggestions:

The paper contains an application of a previous published sensing system to the sensing of temperature. Also a convolutional network (CN) is used to improve the detection resolution. The Cn seems to be the main novelty but is not treated in detail.

Our reply: We thank the reviewer for the comments. We would like to clarify that our work does not utilize a convolutional neural network (CNN) but instead employs a deep neural network (DNN) to improve temperature resolution.

The main contributions of this work include:

  1. On the hardware side, we have demonstrated a clean and cost-effective interrogation method: We propose a practical optical system for interrogating fiber Bragg grating (FBG) temperature sensors, using a waveguide spectral lens (WSL) integrated with a standard CMOS camera. This design eliminates the need for bulky free-space optics and expensive CCD cameras, making the system more compact and affordable.
  2. On the software side, we have achieved much enhanced resolution through DNN: The DNN-based algorithm improves the temperature resolution to 0.1 °C, contributing significantly to the system’s overall performance.

  1. the working of the CN is not adequately described. It generates an improvement of a factor of 70 which needs more justification.

Our reply: We thank the reviewer for the comment. We would like to clarify that we used a DNN rather than a CNN in our study.

We have described the working flow of our DNN-based algorithm in the last paragraph on page 8.

“The spectral image data needs to be preprocessed before feeding into the DNN. As shown in Figure 5(a), the Gaussian filtering is firstly applied on the image to remove high-frequency noise and enhance the contrast of the spectrum line [28]. To ensure the temperature range from 60°C to 120°C fully encompasses the FBG transmission spectrum shift on the camera, we hence select a rectangular region of 2048 × 20 pixels and sum each column to construct a 20 × 1 vector. The vector is determined as the input of the neural network, and the corresponding temperature is selected as the output. The structure of the neural network is also illustrated in Figure 5(a), the number of the layers is set to 4 and the neuron number in each layer is 20, 40, 80, and 1, respectively. An architecture with more layers also works but requires more computation and risks running into overfitting problems, while fewer layers would lead to large errors in the training results and make it difficult for the network to converge.”

To simplify the data processing, the temperature information can be directly demodulated by analyzing the changes in the pixel position of the minimum Vj, which corresponds to the spectral line shift of the FBG center wavelength on the camera. In direct analysis process, the spectral shifts are converted to the displacement change of the valley point of the dark line. However, this displacement should be larger than the width of one pixel on the CMOS camera so that the temperature variations can be distinguished. In our experiments, the CMOS camera has a pixel width of 3.45 mm and the sensitivity are measured as -0.44 mm/°C. The theoretical maximum temperature resolution can only achieve 7.84 °C. The key solution is that the developed DNN-based algorithm considers the location of the valley point in one pixel as well as the relative change between adjacent pixels within overall 20 pixels range. As the dark line shifts from right to left within a single pixel, the intensity changes accordingly. This variation is interpreted as gradient information, reflecting changes in intensity corresponding to temperature shifts. The DNN-based algorithm analyses this gradient information by considering the location of the valley point in each of the 20 pixels. Therefore, the DNN-based algorithm improves the temperature resolution to 0.1°C, representing a more than 70-fold improvement compared to the 7.84°C resolution achieved through the direct analysis method.

We have included additional details in the third paragraph on Page 8 of the revised manuscript, explaining in more detail how the DNN improves the temperature resolution.

“To further improve the resolving ability, we employ DNN-based algorithms to process spectral image data and demodulate temperature information. Within a single pixel, as the dark line shifts from right to left, the intensity changes accordingly. This variation is interpreted as gradient information, reflecting changes in intensity values corresponding to temperature shifts. The DNN-based algorithm analyzes this gradient information by considering the valley point location in each of the 20 pixels and the relative changes between adjacent pixels across the entire 20-pixel range.”

  1. The 2d nature of the (image) appears to be neglected completely despite the fact that a small tilt would allow for sub-pixel resolution. However this appears to be unused for the CN, where only a column average is inserted as input.

Our reply: We appreciate the reviewer's insightful question regarding image tilt and the input to the DNN. We calibrated the image tilt prior to image data collection. The camera used in our experiment has a resolution of 2048×2448 pixels. During the calibration process, we adjusted the tilt angle of the chip or camera and measured the spectrum lines of a single-wavelength laser diode (LD) captured by the camera at different angles. We identified the positions of the maximum pixel values in the first and the 2048th rows of the image. The image is considered leveled when the maximum pixel values for row 1 and row 2048 are aligned in the same column. Therefore, the change in pixel values within each column reflects the change in temperature. We select 20 columns of data and sum the values in each column to form the input for the DNN.

We have included the following text to clarify the image tilt calibration in Section 3, Paragraph 3, on Page 6.

“Similar to our previous work, the wavelength calibration of the WSL was achieved using three laser diodes (LDs). Prior to wavelength calibration, image tilt was calibrated. The camera used in this experiment has a resolution of 2048×2448 pixels. During the calibration process, the tilt angle of the chip or the camera was adjusted, and the spectrum lines of a LD captured at various angles were measured. The positions of the maximum pixel values in the first and 2048th rows of the image were identified. The image was considered leveled when the maximum pixel values in row 1 and row 2048 were aligned in the same column.”

The CN appear to be quite large compared to the small input vector (20 degrees of freedom) and data-set (only 400 for training). It is not clear what distinguishes the validation from the testing data.

Our reply: The WSL-based interrogator utilizes the displacement of the dark line to distinguish temperature, rather than analyzing the entire 2D image. As shown in Figure R1, to ensure that the temperature range of 60 °C to 120 °C fully encompasses the FBG transmission spectrum shift on the camera, we selected a rectangular region of 2048 × 20 pixels and summed each column to construct a 20 × 1 vector (20 degrees of freedom). This vector represents the spectrum displacement information across the 20 columns, allowing the DNN to learn and process the relative changes in intensity between adjacent pixels.

Figure R1. The construction of the input (20 degrees of freedom) for the DNN

During the training process, the mean squared error (MSE) for the training, validation, and testing datasets are all below 2.4 × 10-2 after only 10 epochs, indicating that the model did not overfit during training and the network architecture is appropriate for the input size (20 degrees of freedom) and the dataset (601 images in total). The constructed network effectively captures the relation between the input images and the corresponding temperatures. Furthermore, the validation data was used during the training process to finetune the model's hyperparameters and prevent overfitting, while the testing data was kept separate and only used after the network training was completed to assess the final performance.  

We have added the following description to the revised manuscript on Page 9, in the last paragraph of Section 4.

“A total of 601 spectral images are available for deep learning. Among these images, 60% are randomly selected for training, 20% for validation, and the remaining 20% are used for testing. The validation data was used during the training process to finetune the parameters in the network and prevent overfitting, while the testing data is only used after the model training was completed.”

Reviewer 2 Report

Comments and Suggestions for Authors

Dear colleagues!

Having read your article, I can admit that it is a very clear, methodologically competent article. Maybe I could have missed something, but I am very familiar with such interrogators, and I believe that a high-quality and inexpensive serial product for the considered range of applications can be made on the basis of your research.

The article presents step-by-step the process of manufacturing an interrogator and its application with an analysis at each stage of the advantages and disadvantages of the solutions obtained, and an immediate proposal of solutions to eliminate the latter. At the same time, the range of solutions is very wide from the creation of new spectral elements to neural networks for information processing.

Introduction, conclusion and list of references correspond to and support the significance of the research performed.

I wish you further success!

Author Response

Response Letter to Reviewers

We would like to express our deep gratitude to the reviewers for reading our manuscript carefully as well as for the efforts they have spent in coming up with insightful comments and suggestions to improve the quality of the manuscript. We have tried our best to reply to the comments and made changes in the revised manuscript accordingly.

To make the text clearer and more readable, the original comments from the reviewers and editors are styled italic, our replies are styled partially in bold, and further indications of changes in the manuscript are underlined. The line, paragraph and page numbers are referred to the resubmitted manuscript.

-------------------------------------------------------------------------------------------------------

Reviewer 2

Comments and Suggestions:

Dear colleagues!

Having read your article, I can admit that it is a very clear, methodologically competent article. Maybe I could have missed something, but I am very familiar with such interrogators, and I believe that a high-quality and inexpensive serial product for the considered range of applications can be made on the basis of your research.

The article presents step-by-step the process of manufacturing an interrogator and its application with an analysis at each stage of the advantages and disadvantages of the solutions obtained, and an immediate proposal of solutions to eliminate the latter. At the same time, the range of solutions is very wide from the creation of new spectral elements to neural networks for information processing.

Introduction, conclusion and list of references correspond to and support the significance of the research performed.

I wish you further success!

Our reply: We are very grateful for your kind and positive comments. Thank you for your time and efforts in reviewing our manuscript.